# PIRT the TRP Channel Regulating Protein Binds Calmodulin and Cholesterol-Like Ligands

**DOI:** 10.3390/biom10030478

**Published:** 2020-03-21

**Authors:** Nicholas J. Sisco, Dustin D. Luu, Minjoo Kim, Wade D. Van Horn

**Affiliations:** 1The School of Molecular Sciences, Arizona State University, Tempe, AZ 85287, USA; 2The Virginia G. Piper Biodesign Center for Personalized Diagnostics, Biodesign Institute, Arizona State University, Tempe, AZ 85281, USA

**Keywords:** PIRT, TRP channels, PIP_2_, calmodulin, β-estradiol, nuclear magnetic resonance, microscale thermophoresis

## Abstract

Transient receptor potential (TRP) ion channels are polymodal receptors that have been implicated in a variety of pathophysiologies, including pain, obesity, and cancer. The capsaicin and heat sensor TRPV1, and the menthol and cold sensor TRPM8, have been shown to be modulated by the membrane protein PIRT (Phosphoinositide-interacting regulator of TRP). The emerging mechanism of PIRT-dependent TRPM8 regulation involves a competitive interaction between PIRT and TRPM8 for the activating phosphatidylinositol 4,5-bisphosphate (PIP_2_) lipid. As many PIP_2_ modulated ion channels also interact with calmodulin, we investigated the possible interaction between PIRT and calmodulin. Using microscale thermophoresis (MST), we show that calmodulin binds to the PIRT C-terminal α-helix, which we corroborate with a pull-down experiment, nuclear magnetic resonance-detected binding study, and Rosetta-based computational studies. Furthermore, we identify a cholesterol-recognition amino acid consensus (CRAC) domain in the outer leaflet of the first transmembrane helix of PIRT, and with MST, show that PIRT specifically binds to a number of cholesterol-derivatives. Additional studies identified that PIRT binds to cholecalciferol and oxytocin, which has mechanistic implications for the role of PIRT regulation of additional ion channels. This is the first study to show that PIRT specifically binds to a variety of ligands beyond TRP channels and PIP_2_.

## 1. Introduction

Transient receptor potential (TRP) ion channels respond to a variety of stimuli with pathophysiological implications in pain, obesity, and cancer [1,2,3,4,5,6]. TRPV1 is activated by heat, protons, and capsaicin, which is a pungent vanilloid compound [7,8]. TRPV1 function is further modulated by calcium [9], calmodulin (a calcium-binding protein) [10,11], PIP_2_ [11], and a small membrane protein named PIRT (phosphoinositide interacting regulator of TRPs) [12,13,14]. Functionally opposite to TRPV1, TRPM8 is activated by cold, basic pH, and menthol. Like TRPV1, TRPM8 function is also modulated by calcium [15], calmodulin [11,16,17,18,19,20,21], PIP_2_ [22], and PIRT [23,24]. PIRT modulation of these channels is the least well-characterized.

PIRT, Phosphoinositide-interacting regulator of TRP, was originally identified as a membrane protein that modulates TRP channel function and which binds phosphoinositides [9]. Additionally, it was shown to be expressed in the dorsal root ganglia and trigeminal ganglia of the peripheral nervous system, where, in mice, it enhances TRPV1-dependent currents [12]. More recently, PIRT was shown to regulate TRPV1 with neuropathic pain implications [25], and TRPV1-dependent uterine contraction pain [14]. PIRT is also a regulatory subunit for TRPM8, where it displays species-dependent effects enhancing and attenuating TRPM8-dependent conductance in the rodent and human proteins, respectively [23]. PIRT regulation of TRPM8 appears to arise through competitively binding and modulating access of the lipid PIP_2_ [22,23].

Despite a preponderance of data showing that PIRT regulates TRP channels, there is a growing body of evidence showing that PIRT has regulatory functions that may not be directly related to TRP channels. It has been reported to interact with P2 × 3 channels and that it could impact overactive bladder [26], and also co-localizes with P2X2 within the enteric nervous system [27]. Given that PIRT interacts with PIP_2_, and that a number of PIP_2_-dependent channels are also calmodulin-regulated [11,16,28,29,30], we sought to investigate if PIRT interacts with calmodulin and other common modalities that impact ion channel function.

The highly conserved calcium sensor, calmodulin (CaM), is a known ion channel regulator [17,19,31,32,33] that downregulates Ca^2+^-permeable cation channels, including TRP channels, through a negative feedback mechanism that prevents excessive calcium influx [1]. In TRPV1, CaM competes for a binding site with PIP_2_ and another calcium-binding protein, S100A1 [11]. There is direct evidence that CaM binds to isolated TRPV1 peptides [10]. Complementing this evidence are cryo-EM structures of a related channel, TRPV6, which identifies the location and stoichiometry of the interaction [18]. The structure reveals that CaM binds to TRPV6 in a region that is structurally homologous to the bacterial potassium channel KCNQ1 [21]. It is plausible, based on a conserved CaM binding region in TRPV6, that TRPV1 and TRPM8 bind to CaM in an analogous manner.

TRPM8 is functionally modulated by CaM, where it desensitizes channel conductance in a PIP_2_ dependent manner [16] with calcium influx leading to CaM and PIP_2_-dependent desensitization of TRPM8. It is now understood that TRPM8 binds to and is regulated by both PIRT and PIP_2_ [24,34]; however, it is not currently clear how the CaM-dependent modulatory process functions in the context of PIRT. Nonetheless, with the evidence that PIRT, TRPV1, and TRPM8 bind PIP_2_ [11,12,22,23,24,35,36,37,38], and given the common interdependence of PIP_2_–CaM regulation of ion channels, we hypothesize that PIRT may have a yet to be a determined role in CaM-dependent TRP channel regulation.

Herein, we predict a CaM binding site in the human PIRT using established bioinformatic techniques [39] and then use this information to guide binding assays to identify the amino acids that bind CaM. Our results show that PIRT and CaM bind to each other, which likely leads to the modulation of ion channel access to PIP_2_ and calcium, both of which regulate TRPM8, TRPV1, and other channels with which PIRT may yet interact. Encouraged by the CaM binding data, we used additional bioinformatics tools and surveyed the literature to identify additional potential PIRT binding molecules. Using microscale thermophoresis (MST), our ligand screen focused on a number of cholesterol-like molecules, including cholesteryl-hemisuccinate, cortisol, and β-estradiol, and other TRP channel modulator ligands, including oxytocin and cholecalciferol, we show that the cholesterol-like compounds bind PIRT at a previously unrecognized cholesterol-binding domain (CRAC) [40] and that PIRT shows specificity for β-estradiol but not testosterone and for oxytocin over vasopressin. The ability of PIRT to bind these ligands suggests that it functions as a multimodal TRP channel modulator with implications in other ion channel functions.

## 2. Materials and Methods

### 2.1. Rosetta Flexible Peptide and High-Resolution Docking

We used Rosetta ab initio flexible peptide docking protocols [41] to fold and dock the PIRT C-terminal α-helix (residues 112-137) into the NMR structure of CaM (PDB: 2K0F). We generated 3, 5, and 9-mer fragments for PIRT according to established chemical shift quota filtered fragment generation protocols that are commonly used for Chemical-Shift-Rosetta (CS-Rosetta) [42]. We used these fragments and ensemble state number one from 2K0F to calculate 80,000 docking decoys. The lowest scoring decoy was used to rescore the docked decoys, where the corresponding energy funnel signifies convergence. As per standard docking and Rosetta analysis protocols, we selected the decoy with the lowest interface score decoy with the lowest RMSD as the converged or working model to do a final standard fast relax [43].

### 2.2. Protein Expression and Purification

The expression and purification of PIRT were carried out following the established protocols in our previous work on PIRT [23,24].

Human CaM was expressed at 37 ˚C with a hexahistidine tag comprising MGHHHHHHG- inserted into a pET29 vector with kanamycin resistance and overexpressed in BL21 (DE3) cells. The cells were grown in ^14^N-M9 minimal media (42 mM disodium phosphate (Sigma-Aldrich), 17 mM dipotassium phosphate (Sigma-Aldrich), 8 mM sodium chloride (Sigma-Aldrich), 17 mM ammonium chloride (Sigma-Aldrich), 1× working solution of MEM vitamin solution (Corning), 1 mM magnesium sulfate (Sigma-Aldrich), 100 μM calcium chloride (Sigma-Aldrich), and 22 mM D-glucose (Sigma-Aldrich)). For NMR studies on ^15^N-calmodulin, ^15^N-ammonium chloride was the sole nitrogen source in the M9 minimal media. The cells were induced at OD_600_ = 0.6 with 0.5 mM IPTG (Sigma-Aldrich). The cells were pelleted after 5 hr of induction at 6000×g at 4 °C for 20 min, which resulted in 3 g of cellular mass per 500 mL of M9 culture.

The cell pellet was resuspended and homogenized by tumbling for 1 hr in lysis buffer consisting of lysozyme (0.2 mg/mL), RNase (0.02 mg/mL), DNase (0.02 mg/mL), 1 mM phenylmethanesulfonylfluoride (PMSF, Sigma-Aldrich), 5 mM magnesium acetate (Sigma-Aldrich), 50 mM HEPES (Sigma-Aldrich) at pH 7.7, and 300 mM NaCl. The suspended cells were lysed using a sonicator (QSonica Q500) at 50% duty cycle and 50% amplitude for 7.5 min total on time. Cellular debris was removed with centrifugation of cell lysate at 38,000× *g* at 4 °C for 20 min. The supernatant was used for the following steps after discarding the pellet. The supernatant was tumbled for 1 hr and then loaded onto 2 mL of pre-equilibrated Ni-NTA (QIAGEN: 2 mL of resin per gram of cell pellet) within a gravity column. The Ni-NTA was pre-equilibrated with lysis buffer (50 mM HEPES (Sigma-Aldrich) pH 7.7 and 300 mM NaCl). Purification was then carried out with a flow-through buffer (50 mM HEPES pH 7.5), low imidazole wash (50 mM HEPES pH 7.5, 10 mM imidazole), and finally an elution buffer (50 mM HEPES pH 7.5, 300 mM imidazole). Following Ni-NTA chromatography, the eluent was concentrated to a volume of 500 μL and loaded directly onto a 60 mL column volume Superdex 200 (GE Healthcare Life Sciences) pre-equilibrated with 50 mM HEPES at pH 7.5 and separated by size. CaM purity was assessed with SDS-PAGE and the identity confirmed by western blot analysis with a penta-histidine primary antibody and anti-mouse alkaline phosphatase detection (Appendix A). We then used NMR to show that our CaM construct retains functional properties by heteronuclear single-quantum coherence (HSQC) NMR in the presence and absence of calcium. The resulting spectra show characteristic chemical shift perturbation for CaM (Appendix A).

For the pull-down experiment, the His-tagged CaM was buffer exchanged using a 10 kDa cutoff Amicon Ultra centrifugal filter (Millipore) into a thrombin cleavage buffer (25 mM Na_2_HPO_4_, 150 mM NaCl, pH 7.8) following Ni-NTA purification. The sample was then tumbled with 3 units of thrombin (Novagen) for 24 h at room temperature before flowing over Ni-NTA. The collected flow-through contained His-tag cleaved CaM, which was further purified by gel filtration chromatography.

### 2.3. Microscale Thermophoresis

Human PIRT was fluorescently labeled according to previous protocols [24]. For the MST measurements, the concentration of PIRT was kept consistent at 200 nM for all ligands in MST buffer (0.1% DPC (*w/v*), 50 mM HEPES, pH 7.0). Apo-CaM was purified as mentioned above, with 0.5 mM EDTA added to the size exclusion buffer and kept consistent throughout the MST measurements.

Cortisol, β-estradiol, testosterone, cholesteryl-hemisuccinate, and cholecalciferol were all solubilized as stock solutions in neat chloroform. From the ligand stock solutions, the desired amount of the ligand for MST experiments was aliquoted into 200 μL PCR tubes, and the chloroform was evaporated off under streaming N_2_ at room temperature. The compounds were then resuspended in MST buffer with 200 nM PIRT. The disposal of testosterone was carried out according to U.S. Drug Enforcement Administration (DEA) standards under Title 21 Code of Federal Regulations. Oxytocin, vasopressin, calcium chloride, and nicotinamide stock solutions were prepared directly in MST buffer and aliquoted to desired concentrations for MST analysis.

The MST labeled PIRT was added to the solutions and incubated for 1 hr at room temperature. After incubation, a standard MST glass capillary tube (NanoTemper) was drawn into the tube by capillary action (~5 μL). MST experiments were carried out in triplicate at room temperature with 50% infrared laser power and green channel using 10% excitation power.

All of the data from the MST ligand screen that showed ligand-dependent thermophoresis were normalized to free PIRT and bound PIRT following established protocols [44,45], from which the dissociation constant was calculated. Ligand-independent (i.e., non-binding) thermophoresis is evident in the controls (Appendix A). Ligand titration concentrations were optimized to show saturation and minima within the bounds of solubility; i.e., the cholesterol-like ligands tended to become insoluble above the concentrations used. The data were fit with in-house Python scripts where the errors are reported as the root-mean-square error (RMSE) of the fit. Typically, membrane protein binding studies of hydrophobic compounds use units of mole percent [24,44,46]. However, given that we screened both hydrophobic and hydrophilic compounds, we used molar-dependent units with a constant PIRT and membrane mimic concentration in order to compare the binding studies across chemical environments.

### 2.4. PIRT and Calmodulin Pull-Down Assay

The pull-down experiment was conducted by mixing 10 μg His-tagged PIRT to 20 μg CaM (sans His-tag) and tumbled at room temperature to allow the mixture to come to equilibrium. The sample was then exposed to 200 μL of Ni^2+^-bound nitrilotriacetic acid (Ni-NTA) resin and washed with 50 column volumes of 50 mM HEPES, 0.1% DDM, pH 7.5 to elute any unbound CaM. The bound sample was then eluted with 50 mM HEPES, 0.1% DDM, 500 mM imidazole, pH 7.5, and analyzed by SDS-PAGE gel.

### 2.5. Nuclear Magnetic Resonance-Detected Binding Assay

^15^N-human PIRT (180 μL, 3 mm NMR tube) and ^15^N-human CaM (550 μL, 5 mm NMR tube) were measured in NMR buffer (4% D_2_O (v/v, Sigma Aldrich), 20 mM sodium phosphate (Fisher Scientific), 500 µM DSS (Sodium trimethylsilylpropanesulfonate, Sigma Aldrich), and 0.5 mM EDTA (Sigma Aldrich) at pH 6.5) on a Bruker 850 MHz ^1^H magnet with Avance III console. Two thousand forty-eight direct points and 128 indirect points were collected with 128 transients, processed in NMRpipe [47], and analyzed in CCPNMR [48]. Optimization of NMR conditions for PIRT was carried out previously [24], resulting in dodecylphosphocholine (DPC) as the most suitable detergent for investigations with PIRT and at a temperature of 40 °C. An HSQC of ^15^N-human CaM was measured with and without CaCl_2_ to show that it was properly folded in the conditions we tested.

To validate the cholesterol-like and β-estradiol binding site, a saturating concentration of β-estradiol (3.82 mole %) was dissolved in DMSO before adding to a ^15^N-human PIRT sample. An HSQC of human PIRT was measured before and after adding β-estradiol, and the resonances with significant chemical shift perturbation identify the amino acids that comprise the binding site.

## 3. Results

### 3.1. Bioinformatic Analysis of PIRT Predicts Calmodulin-Binding Motifs

The CaM 1-14 motif has a consensus sequence of (FLIVW)-X_12_-(FLIVW), where the X is any amino acid flanked by hydrophobic Phe, Leu, Ile, Val, or Trp. With the CaM target database web server [49], PIRT was identified to contain a highly conserved 1-14 calmodulin-binding motif in the C-terminal -helix, and there are two possible motif positions located at residues Ile113 or Ile114 to Phe126 or Leu127 (Figure 1 and Appendix A). Residue number 127 is not 100% conserved for a leucine; however, sequence homology shows that phenylalanine can occupy position 127, which is still consistent with a 1–14 motif. Worth noting is that PIRT may have an additional CaM motif of 1–16 from position Val111 to Phe126 (Figure 1A). However, if this motif location truly binds PIRT, we are unable to resolve it with our available NMR data.

Using the GPS 2.0 tool, we identified two previously unknown calmodulin-dependent kinase II (CaMKII) motifs with the sequence R-X-X-S/T in PIRT, both of which have very high conservation (Appendix A) [50]. The amino acids Arg36 to Ser39 are 100% conserved across the phosphoinositide interaction proteins and are consistent with the minimal recognition site for CaMKII [50]. There is less conservation for the second CaMKII site at amino acid locations Arg121 to Ser124, but conservation is still high with Arg121 to being overall highly conserved. We did not attempt to validate the predicted CaMKII motifs, but 185 mention them here for completeness.

### 3.2. Flex Pep Dock

Using Rosetta flexible peptide ab initio docking protocols, we were able to model how the positively charged C-terminal α-helix of PIRT might bind with the negatively charged calmodulin. We previously assigned the amino acid backbone resonances for the C-terminal amino acids from Lys117–Arg137 and used these assignments to make experimentally constrained fragments, which take into account the φ/ψ angles derived from NMR as well as evolutionarily conserved φ/ψ angles for stretches of similar amino acids. The Rosetta calculations show clear convergence (Appendix A) and localize the C-terminal α-helix to our predicted 1–14 CaM-binding motif (Figure 1A). The helical wheel representation (Figure 1B), made with using the European Molecular Biology Open Software Suite [51], shows that it is certainly possible for there to be two different 1–14 site that would be amenable for binding within the CaM hydrophobic pockets; however, our models predominately found the position Ile114 to Leu127 to be the lowest-scoring; i.e., lowest energy, and therefore, best docking (Appendix A).

### 3.3. PIRT Specifically Interacts with Calmodulin

PIRT has a high affinity for calcium-free CaM (apo-CaM) and a reduced affinity for calcium bound calmodulin. The MST binding screen shows that PIRT binds tightly in the mid to low nM range, *K_d_* = 350 ± 40 nM for apo-CaM (Table 1, Figure 2A). Calcium bound CaM (holo-CaM) has a decreased affinity for PIRT(i.e., weaker binding) and right-shifts the binding curve by about 200-fold to an apparent *K_d_* = 60 ± 30 μM (Table 1, Figure 2A). PIRT does not bind to calcium chloride (Appendix A).

To verify the CaM–PIRT interaction, a pull-down assay was conducted by mixing purified His-tagged PIRT with His-cleaved CaM at room temperature and then tumbled with Ni-NTA resin. The sample was washed extensively with 50 column volumes to remove any unbound protein. CaM co-elutes with the imidazole elution of PIRT. The Coomassie stain gel shows that PIRT and CaM were co-eluted confirming direct binding between PIRT and CaM (Appendix A).

### 3.4. NMR Titration Shows Residues with Ligand-Dependent Perturbation Including the Predicted Calmodulin Binding Site

NMR-detected ^15^N labeled PIRT titrations with CaM show chemical shift perturbations for calcium-free CaM (Figure 3A) within the C-terminal α-helix (Figure 3B). Interestingly, Ser124 shows perturbations suggestive of slow-exchange on the NMR timescale with the disappearance of the initial peak, presumably the ligand-free state, and reappearance of the apparent bound peak across the titration concentrations; a hallmark of slow-exchange and tight binding. In addition to Ser124, chemical shift perturbation is also seen for His120 in the predicted 1–14 calmodulin-binding motif, while residues Lys131, Ser132, Leu135, and Arg137 also show chemical shift perturbation and are found in close proximity to the binding motif (Figure 1C).

### 3.5. Two Cholesterol-Binding Motifs Found on PIRT with Bioinformatic Analysis

PIRT has two putative cholesterol binding motifs in the first transmembrane α-helix. Cholesterol-binding proteins share a consensus sequence motif named cholesterol-recognition amino acid consensus called CRAC, and CARC, where CARC is the mirror image of CRAC. The CRAC amino acid sequence is (L/V)-X_1-5_-(Y)-X_1-5_-(K/R), with N to C terminal directionality, and X stands for any amino acids [40]. Using the known molecular motifs for cholesterol-binding proteins, we report two cholesterol-like binding sites in the first transmembrane α-helix of human PIRT with CRAC in the outer leaflet of the membrane bilayers and CARC in the inner leaflet (Figure 4 and Appendix A).

### 3.6. PIRT and Cholesterol Ligands Screening with Microscale Thermophoresis

We used microscale thermophoresis to investigate PIRT–cholesterol interactions. Given the solubility limitations of cholesterol, we tested the predicted cholesterol-binding with the more soluble cholesteryl-hemisuccinate, which shows an affinity of 103 ± 6 μM (Figure 4). Cholesterol derivatives also bind PIRT with cortisol and β-estradiol K_d_ values of 790 ± 60 μM and 800 ± 100 μM, respectively (Figure 4). Surprisingly, testosterone does not bind to PIRT (Appendix A) despite its structural similarity to cholesterol, cortisol, and β-estradiol.

### 3.7. β-Estradiol Induced PIRT Chemical Shift Perturbation

To validate the predicted CRAC cholesterol-binding site in PIRT, a saturating concentration of β-estradiol was added to ^15^N labeled PIRT. The resulting TROSY-HSQC spectra show chemical shift perturbations primarily in the upper transmembrane α-helix 1 (TM1) with a few that breach to the upper transmembrane α-helix 2 (TM2) (Figure 5). More specifically, chemical shift perturbations were seen at residues Ser60, Val61, Thr73, Tyr77, Lys80, and Leu81 in TM1 while Gly95 and Leu97 are seen in the TM2. The NMR data confirm the CRAC binding site location (Figure 4 and Figure 5). The NMR data also show limited perturbation away from the CRAC site upon β-estradiol binding, with chemical shift perturbation of Ser60 and Val61 in TM1 and Gly95 and Leu97 in TM2, which are outside of the predicted CRAC domain, this could indicate that β-estradiol induces a change in PIRT conformation or dynamics.

### 3.8. Microscale Thermophoresis Shows PIRT Binds to Other Trp Channel Modulators

Cholecalciferol, the pro-hormone form of vitamin-D synthesized in the skin, has low structural similarity to cholesterol-like molecules but is known to affect TRPV6 [52]. The MST data show that PIRT specifically interacts, albeit with low affinity, to this secosteroid with an apparent *K_d_* = 2.1 ± 0.4 mM (Appendix A). Similarly, PIRT affects oxytocin-induced uterine contraction pain in a TRPV1-dependent manner [14], which we used to test the hypothesis that PIRT may bind to oxytocin. Remarkably, PIRT binds to oxytocin with a *K_d_* = 7 ± 1 μM (Appendix A), and yet it does not bind to the structurally similar peptide hormone R8-arginine vasopressin (Appendix A).

To show the significance of the MST binding data, we used nicotinamide (Appendix A) as a control because it has well-known metabolism and should not interact with PIRT. In previous MST PIRT binding studies using similar conditions, DMSO was previously shown to not bind to PIRT [24].

## 4. Discussion

Regulation of TRP channels by PIP_2_, Ca^2+^, and CaM is an active area of research. Several reviews highlight the important features of these TRP channel modulatory mechanisms [53,54,55,56,57]. For some TRPV channels, it is clear that CaM directly binds these channels and regulates function [58,59]. Structural studies of TRPV5 indicate that a single CaM can bridge two distinct TRPV5 binding sites [58]. Similarly, TRPV1 has at least two CaM binding sites, one in the N-terminal ankyrin repeat domain (ARD) and another in the distal C-terminus (CT) [59]. The ARD CaM-binding site functions in desensitization, while the CT CaM interaction has a higher affinity (lower *K_d_*), it has a less defined functional consequence [59]. Titration of holo-CaM to the isolated TRPV1 distal C-terminus (CT) as monitored by tryptophan fluorescence emission identifies an affinity of *K_d_* = 5.4 × 10^−8^ M [59] which is on par with the affinities identified between PIRT and CaM in this study. However, one significant distinction is that PIRT has a higher affinity for apo-CaM; whereas, it is generally thought that holo-CaM, the Ca^2+^-bound form, is the regulatory form that plays a role in TRP channel desensitization [55]. While CaM modulation can be direct (i.e., direct binding to TRP channels), modulation can also happen indirectly by impacting other TRP regulatory proteins such as CaM-dependent regulation of phosphatases or kinases [16]. In TRPM8, CaM has been clearly shown to downregulate function indirectly [16]. There is also some evidence that CaM directly binds to TRPM8 [60,61], which would provide an additional mechanism of CaM-mediated functional downregulation. Beyond Ca^2+^–CaM modulation, at least for TRPM8 and other TRPM channels, it is clear that Ca^2+^ can directly bind and desensitize these channels in the absence of CaM [62]. Our PIRT studies identify a CaM-binding site that adds an additional layer of complexity to TRP channel function and modulation, where PIRT could impact both direct and indirect CaM regulatory functions.

Using purified full-length human PIRT, we show it binds to CaM at the predicted C-terminal α-helix, which suggests a possible role in regulating TRP channel function in concert with calcium and PIP_2_ and suggests a broader modulatory role in ion channel regulation. With MST, calcium bound CaM showed an approximately 200× decrease in affinity for PIRT than for the apo (calcium unbound) form, which likely arises from known CaM conformational changes upon calcium loading and suggests that calcium-binding of CaM may induce the dissociation of the PIRT–CaM interaction (Figure 2). The PIRT–CaM interaction is confirmed with a pull-down experiment showing that CaM co-eluted with PIRT elution.

We further tested PIRT binding to CaM with NMR titrations to provide amino acids specific details and validate our theoretical CaM binding site on PIRT. Using our previously published PIRT amino acid resonance assignments [24], we show that calcium-free CaM binds to PIRT in the predicted C-terminal α-helix and that the changes in chemical shift indeed correspond to the predicted CaM-binding motif (Figure 1C). The CaM-dependent chemical shift perturbations show several amino acids that display binding, highlighting a possible allosteric effect arising from the interaction. To model our biochemical and biophysical measurements, we used the chemical shift assignments to make Rosetta fragments allowing us to computationally model CaM bound to the PIRT C-terminal α-helix. Our docked model shows that the C-terminal α-helix fits well within the conserved CaM-binding motif with the hydrophobic Ile113 fitting in the C-terminal lobe and Leu127 in the N-terminal lobe of CaM from PDB entry 2K0F (Figure 1).

Apo-CaM binding to PIRT can be used to expand on known CaM downregulation of the TRPM8 function [16]. Cryo-EM structures of TRPM8 show that Ca^2+^ binds to the human TRPM8 S1-S4 membrane domain with Ca^2+^ chelated by Glu782, Gln785, Tyr793, Asn799, and Asp802 to prime the channel for activation [34,62]. These amino acids, as well as Glu1068, are conserved for all Ca^2+^ dependent TRPM channels (TRPM2, M4, M5, and M8) [15,63,64,65]. This conservation for Ca^2+^ dependent TRP channels, and their structural homology, suggest that TRPM8 potentially binds to Ca^2+^ in the same location and potentially has a similar effect on the channel. In TRPM8 channels, especially, elevated levels of intracellular Ca^2+^ causes CaM downregulation of TRPM8 with a mechanism that depends on PIP_2_ [16]. We previously showed that PIRT reduces the human TRPM8-dependent currents by binding directly to the human TRPM8 S1-S4 domain [23] and shuttles PIP_2_ to TRPM8 [24], and this mechanism is plausible for PIP_2_ specific inactivation of TRPM8. The data presented here support a more complex mechanism that contextualizes CaM, PIP_2_, Ca^2+^, and PIRT downregulation of TRPM8 dependent currents.

Figure 6 contextualizes the known interactions that have been identified and relate CaM, Ca^2+^, PIRT, and PIP_2_ to TRPM8 modulation. TRPM8 integrates a variety of stimuli, including, but not limited to, temperature, menthol, and pH. These inputs are then modulated by an additional layer of interactions between PIRT, CaM, and PIP_2_, where PIRT can bind to apo-CaM and PIP_2_, thereby modulating TRPM8 access to these modulators. Currently, the regulatory cross-talk between CaM and PIP_2_ is unknown, but given the spatial locations and apparent overlap of the respective PIRT binding sites, it likely impacts the regulatory network. TRPM8 gating by its canonical stimuli (i.e., cold) is thereby further tuned by a variety of modulators, including PIRT, that will ultimately regulate calcium influx and thereby initiating signal transduction. Increases in intracellular calcium concentrations result in direct Ca^2+^-dependent TRPM8 desensitization [62]. Increased available calcium will also impact CaM leading to higher concentrations of holo-CaM. As a result, because of the decreased affinity of the PIRT—holo-CaM complex, we speculate that the complex would dissociate. Once the calcium levels return to equilibrium, then the PIRT—CaM interaction cycle would continue with apo-CaM rebinding PIRT. The apo-CaM and PIRT complex could potentially allow the channel to become active again by allowing PIP_2_ to be shuttled to TRPM8 when the complex is formed. More studies that test the intricacies of these putative interactions are still needed, but our data support a more detailed negative regulation from Ca^2+^ for TRPM8-dependent currents.

PIRT appears to regulate function via diverse means. In this vein, we identified a conserved cholesterol-binding domain called CRAC [40] in the first transmembrane α-helix of PIRT, and we show that it does indeed bind to several cholesterol-like molecules with potential implication in ion channel regulation [66]. We chose these ligands based on bioinformatics predictions on the PIRT sequence matching cholesterol-binding motifs as well as ligands implicated in uterine contraction pain (oxytocin and β-estradiol) and a pro-hormone form of vitamin D with ties to TRPV6 (cholecalciferol) [67]. Our data is the first to show that PIRT binds to cholesterol-like molecules with specificity for cholesteryl-hemisuccinate, cortisol, and β-estradiol; however, it does not specifically bind testosterone. Its interaction with cortisol suggests that it may play a more intimate role in TRP channel-specific stress-induced inflammatory processes.

Binding to cholesteryl-hemisuccinate supports cellular electrophysiology measurements where PIRT was shown to interact with the TRPM8 pore domain [23], and it is known that TRPM8 interacts with cholesterol to partition it into cholesterol-rich membrane domains [68]. Additionally, TRPA1 also has a CRAC domain, which affects both membrane partitioning and function [69]. Furthermore, cholesteryl-hemisuccinate was bound in cryo-EM structures of a handful of TRP channels, including TRPC4, TRPM4, TRPV3, and TRPM8 [62,63,70,71]. With this physiological and structural data, and our cholesterol-binding data and bioinformatics, we have uncovered another layer of PIRT regulation that may implicate it in is much more diverse physiology than previously thought.

Interestingly, while β-estradiol (female sex hormone) binds tightly to PIRT, testosterone (male sex hormone) shows no evidence of binding (Appendix A). This suggests that there may be a sex-specific regulatory role in which PIRT plays a part and may partially explain the role PIRT plays in uterine contraction pain [14]. In addition to uterine contraction pain, TRP channels have been implicated to play a role in cancer cells where expression of these channels can affect cancer cell proliferation, metastasis, and even cell death [72]. In breast cancer cells, the TRPV family and TRPM8 channels expression are regulated by β-estradiol and estrogen receptors [73,74]. Our MST and NMR data show that PIRT binds tightly to β-estradiol and induce some evidence of conformational change in the HSQC, which suggests PIRT might play a role between TRP channels with β-estradiol in breast cancer cells. Supporting the role of PIRT in sex-dependent function, a recent study of PIRT knock-out (^-/-^) mice indicate that there are subtle PIRT-dependent impacts in metabolism and obesity that are female-specific [75].

Our ligand screen shows that PIRT binding is not limited to cholesterol-like hormones and that cholecalciferol (Vitamin D3) specifically interacts with PIRT. Cholecalciferol is a hormone synthesized in the skin from a cholesterol derivative but is structurally distinct from cholesterol with the sterol backbone disrupted from UV radiation provided by the sun. Vitamin D3 is well known to be intimately involved in bone formation, with recent research on dietary calcium and vitamin D3 being shown to regulate the epithelial ion channels TRPV5 and TRPV6 [67]. It is unknown what role PIRT would directly have on TRPV6 and TRPV5; however, based on emerging evidence of PIRT as a general TRP channel regulator, it is plausible that PIRT regulates TRPV5 and TRPV6 in concert with the calcium sensor CaM.

Lastly, our studies indicate that PIRT interacts specifically with peptide hormones. We show that PIRT specifically binds oxytocin. It is plausible that PIRT is involved with oxytocin regulation given that the oxytocin receptor is a G_qα_ type G protein-coupled receptor (GPCR), G_qα_ GPCRs directly influence PIP_2_ levels, GPCRs are implicated with TRP channels, and PIRT is a PIP_2_ binding protein that functionally modulates TRP channels [12,13,14,23,25,76,77,78]. Similarly, PIRT was shown to be important for uterine contraction pain under oxytocin insult during birth in mice [14]. Interestingly, while PIRT specifically interacts with oxytocin, it does not bind R8-arginine vasopressin, despite only two amino acid differences in the structurally similar cyclic nonapeptides. It is notable that these hormones have drastically distinct roles in endocrinology, controlling birth induction and blood pressure regulation, respectively. In fact, oxytocin-induced childbirth can result in off-target effects complicating childbirth that are thought to be caused by the similarity of these hormone structures [79,80]. These effects are not fully understood, but with the identification of PIRT, specifically interacting with these hormones, potentially new avenues to understand these side effects can be investigated.

## 5. Conclusions

The data presented here show several binding partners for PIRT with specific endocrinology roles as well as implicate an enhanced model for TRPM8 regulation. These are the first data to suggest PIRT interactions beyond those with TRP channels and PIP_2_, which opens up multiple avenues of PIRT-related research.

## Figures and Tables

**Figure 1 biomolecules-10-00478-f001:**
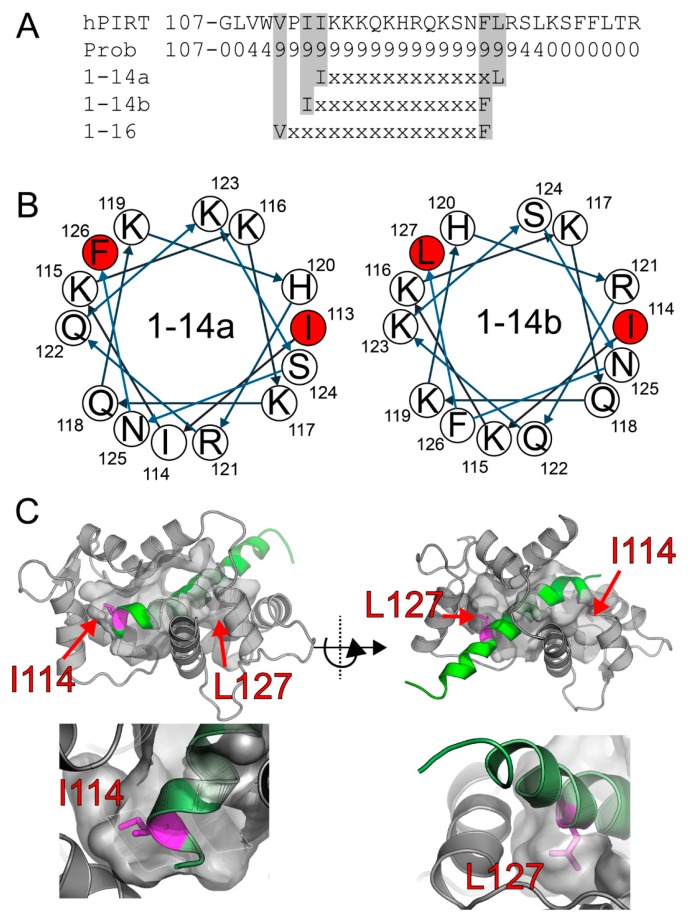
PIRT has a highly conserved calmodulin-binding motif found in the C-terminal α-helix. (**A**) We performed bioinformatic analysis on the PIRT sequence and show here that it has putative 1–14 and 1–16 motifs. We exclude the 1–16 motif as Val111 is likely found well within the membrane bilayer, preventing calmodulin-binding. The helical projections in (**B**) show that both 1–14 motifs are consistent with an α-helix with hydrophobic residues that would occupy the calmodulin-binding pockets in the N- and C-terminal lobes. (**C**) Using Rosetta ab initio flexible peptide docking, we folded the C-terminal α-helix into the calmodulin binding pocket and highlight that the peptide that fits best is the 1–14 motif with Ile114 and Leu127.

**Figure 2 biomolecules-10-00478-f002:**
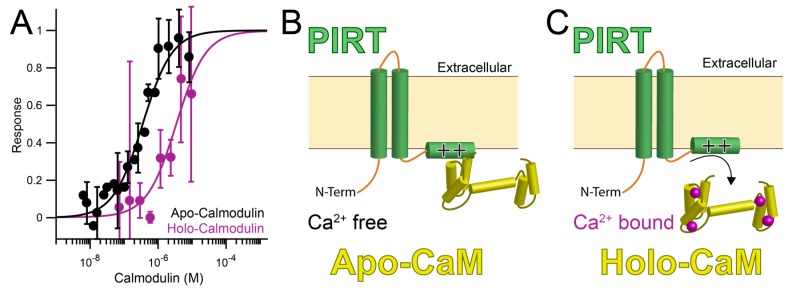
PIRT (Phosphoinositide-interacting regulator of TRP) binds to calcium-free calmodulin (apo-CaM) and calcium bound calmodulin (holo-CaM). (**A**) PIRT binds calcium-free calmodulin (apo-CaM, black) with approximately 200-fold higher affinity than Ca^2+^-bound calmodulin (holo-CaM, purple). (**B**) The differences in affinity suggest the possibility that PIRT is bound to apo-CaM until intracellular calcium levels rise high enough for holo-CaM to bind calcium and release PIRT (**C**).

**Figure 3 biomolecules-10-00478-f003:**
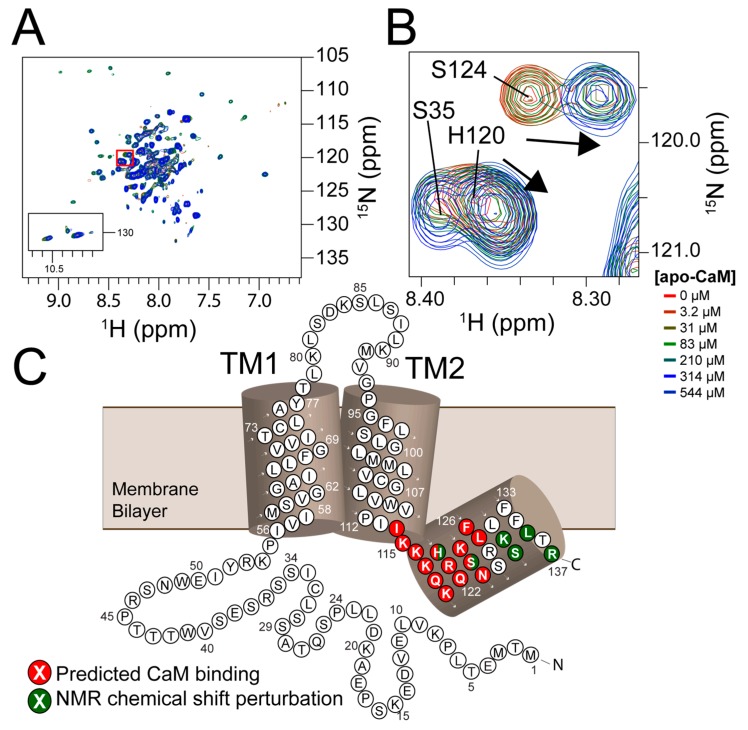
PIRT has residues that bind to calcium-free calmodulin in the C-terminal α-helix that is located in or near the 1‒14 motif. (**A**) The HSQC follows the titration of NMR invisible ^14^N-calcium-free calmodulin with NMR-detected ^15^N-PIRT with concentrations of 0, 3.2, 31, 83, 210, 314, 544 μM calcium-free calmodulin. (**B**) Highlighted from the red inset on the HSQC are Ser35 showing no perturbation, His120 showing chemical shift perturbation show with the arrow, and Ser124 showing an apparent slow exchanging resonance with the arrow highlighting the movement. In (**C**), the calmodulin-binding motif is highlighted in red from Ile114 to Leu127 that was shown with Rosetta flexible docking to be the best fitting motif. Highlighted in dark green are residues that show chemical shift perturbations with the NMR titration.

**Figure 4 biomolecules-10-00478-f004:**
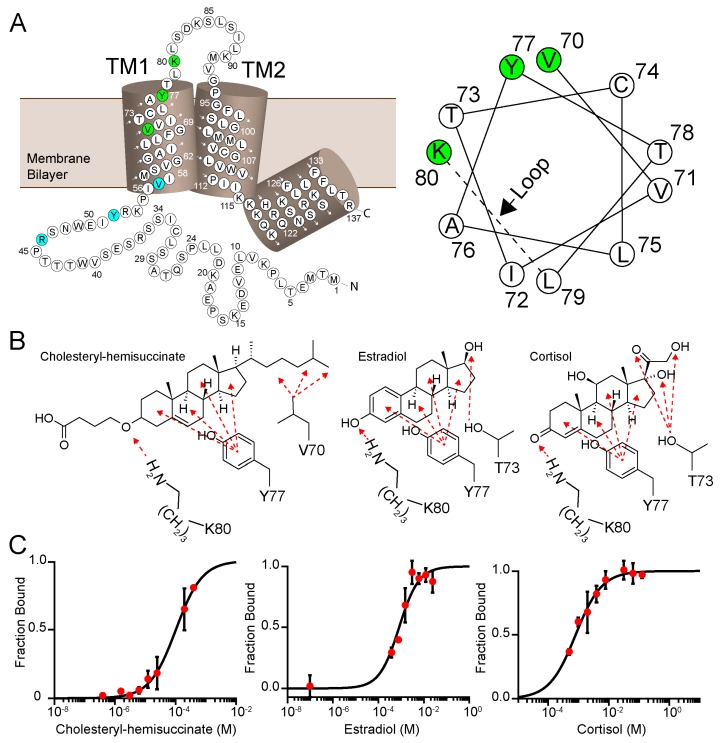
PIRT has a conserved cholesterol-binding CRAC and CARC motif. (**A**) We analyzed the PIRT sequence and identify that it contains a CRAC motif in the outer leaflet of the first transmembrane α-helix with amino acids highlighted in green and a CARC domain in the inner leaflet with amino acids highlighted in cyan. Shown in the helical wheel projection, the CRAC domain is supported with the evidence that within a standard α-helix, the Val70-Tyr77-Lys80 are on the same interface and are confirmed to bind to cholesterol with the NMR data below. (**B**) The amino acids that may bind to cholesterol hemisuccinate, β-estradiol, and cortisol and are shown here as an example of how PIRT could bind these ligands. To bind the β-estradiol or cortisol, we show here that PIRT could incorporate Thr73, which is on the same interface as the CRAC domain residues. In Appendix A, our alignment highlights a second cholesterol-binding domain, CARC, it resides outside of the membrane bilayer, does not have a helical interface consistent with a standard α-helix, and is not highlighted here. (**C**) PIRT ligand-dependent thermophoresis of cholesteryl-hemisuccinate, β-estradiol, and cortisol. The *K_d_* values are listed in Table 1.

**Figure 5 biomolecules-10-00478-f005:**
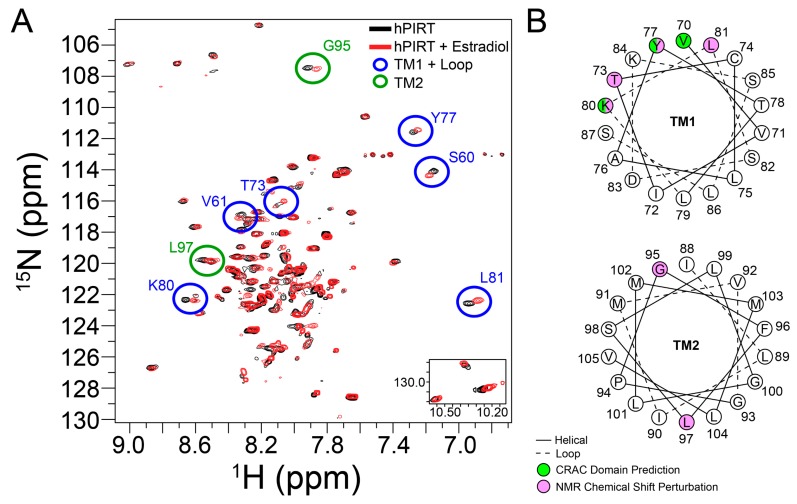
Identification of the PIRT-β-estradiol binding site by NMR. (**A**) The NMR HSQC of PIRT (black) overlaid against PIRT with a 3.82-mole percent β-estradiol (red). Chemical shift perturbations are seen in the first transmembrane α-helix (S60, V61, T73, Y77, K80, and L81) and second transmembrane α-helix (G95 and L97). (**B**) Helical wheel projection of the first and second transmembrane α-helices (TM1 and TM2, respectively) with the predicted CRAC domain (green) compared to the NMR chemical shift perturbation (pink). The chemical shift perturbation seen in the upper sections of the α-helices supports the CRAC domain prediction. Residues S60 and V61 in TM1 and G95 and L97 in TM2 suggest there might be a conformational change when β-estradiol binds with PIRT.

**Figure 6 biomolecules-10-00478-f006:**
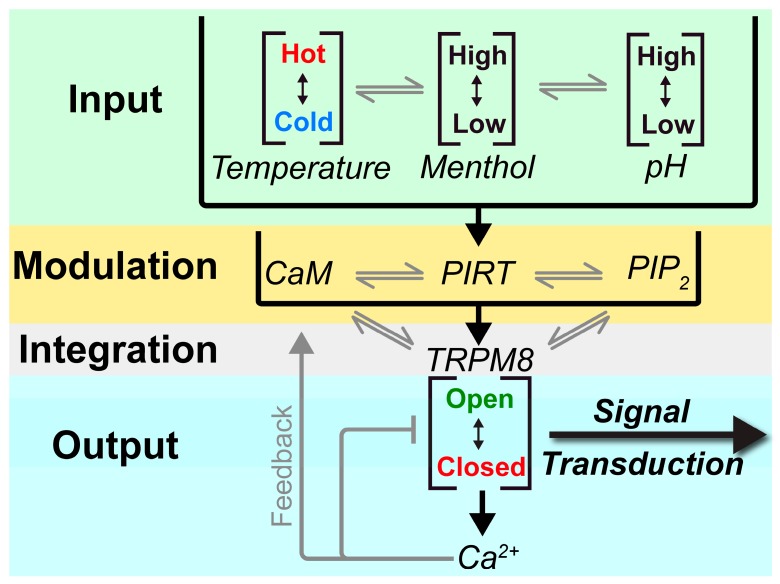
A diagram relaying the interactions known so far between CaM, PIRT, PIP_2_, TRPM8, and Ca^2+^. Inputs are taken and are modulated by the interactions between CaM, with PIRT and PIRT with PIP_2_. These interactions are then integrated to TRPM8, where the channel responds by either opening or closing. Once the channels open, calcium influx occurs, causing signal transduction, and the influx of calcium causes a feedback loop where calcium will bind CaM while also sending negative feedback to TRPM8 and desensitize it.

**Table 1 biomolecules-10-00478-t001:** Ligands bound to PIRT using MST.

Ligand	K_d_ (M)	RMSE (M)	Ligand Type
Calmodulin (apo, free)	350 × 10^−9^	40 × 10^−9^	CaM, Intracellular Protein
Calmodulin (holo, bound)	60 × 10^−6^	30 × 10^−6^	Calcium Bound CaM
Cholesteryl HS	103 × 10^−6^	6 × 10^−6^	Steroid Hormone Precursor
Cortisol	790 × 10^−6^	50 × 10^−6^	Cortical Steroid Hormone
β-Estradiol	800 × 10^−6^	100 × 10^−6^	Sex Steroid Hormone
Cholecalciferol	2.1 × 10^−3^	0.4 × 10^−3^	Secosteroid Hormone
Oxytocin	7 × 10^−6^	1 × 10^−6^	Peptide Hormone

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
