# Peer review of "PIRT the TRP Channel Regulating Protein Binds Calmodulin and Cholesterol-Like Ligands"

_biomolecules, 2020, doi:10.3390/biom10030478_

Round 1

Reviewer 1 Report

General comments: The authors present a compelling story wherein PIRT and CaM interact to modulate TRP channels. The manuscript is focused on identifying and then confirming the interaction between PIRT and CaM. It also evaluates another binding domain on PIRT, the CRAC domain, by testing several cholesterol derivatives. Nice data in Figure 5!

Here are some suggestions to improve the manuscript. On line 168, supplemental figure 3 is references in the context of the web server database however the figure is thermophoresis data. Perhaps this is a typo?

The error bars in Figure 3A and B (particularly in B) are some cause for concern. It would be useful if the authors could address this. The lack of a solid plateau on either side of the curve makes the error bars even more important. This does not bring into question the result that holo-CaM binding with lower affinity though.

The Rosetta modeling was done with a calcium bound and peptide bound structure of CaM. The authors should address how they think this will influence the model particularly in light of their own data showing higher affinity binding with apo-CaM.

The pull down assay was done with His-PIRT and a non-His-tagged CaM, however the methods does not mention a purification of a wild type CaM nor a cleavage of the His tag from the His-CaM. Please include either the purification or the cleavage conditions.

In Figure 3, several PIRT concentrations are listed in the legend however it is unclear in the figure, which concentration corresponds to which data. A color coded legend would be required here.

While the cartoons and structures of Figure 4 are useful, the main point is the binding the cholesterol derivatives to PIRT. Some of the data from supplemental Figure 7 should be included in Figure 4 in the main text so the reader can easily see the binding that is discussed. Also in Figure 4, most of the binding curves do not have solid plateaus. Are these data limited by the technique or the solubility of compounds? Some explanation would be appreciated here.

Language issues:

Line 72 “literature to suggest”.

Line 292: “increase in calcium in the inside of the cell causes….”

Line 296: “shuttled

Line 315: “in cryo-EM structures of a handfull….”

Reviewer 2 Report

In the manuscript the authors describe many novel interaction partners of PIRT, which leads to the impression that PIRT is not only a general TRP channel modulator, but also affects other ion channels. The described results may help to elucidate the complex regulation mechanism of Trp channels.
To investigate binding the authors used mainly prediction tools, MsT and NMR methods. They also provide an interesting model how CaM PIRT may modulate Trp channels.
But there are a few major and minor improvements to be performed:
A) PIRT- Calmodulin interaction
minor
1) concerning the mst measurements: Fig. 2A and B could be integrated, so the shift is seen more clearly. Why the measurements with holo CaM show so large variations. Did you provide a bonding check. Are triplicates really enough?
2) Please provide the complete SDS gel ( Fig. S6)
3)There is also a direct Cam binding reported to the Trp channels. How does this fit to your model . Please address to this in your discussion
4)Line 72 "encouraged by.. " and line 196 " In the presence of saturation calcium, load CaM,...." please reconstruct these sentences. The meaning is not clear
B) PIRT and Cholesterol derivatives
2)line 330-337 Do the authors suggest that PIRT+Vit D3+ CaM regulate TRPv5/6 in concert the connection is not very clear. The binding of vit D3 is in mM range, thus very weak.
